# Preparation and Characterization of Chitosan/LDH Composite Membranes for Drug Delivery Application

**DOI:** 10.3390/membranes13020179

**Published:** 2023-02-01

**Authors:** Elena-Ruxandra Radu, Andreea Madalina Pandele, Cristina Tuncel, Florin Miculescu, Stefan Ioan Voicu

**Affiliations:** 1Department of Analytical Chemistry and Environmental Engineering, Faculty of Chemical Engineering and Biotechnologies University Politehnica of Bucharest, 011061 Bucharest, Romania; 2Advanced Polymers Materials Group, University Politehnica of Bucharest, 011061 Bucharest, Romania; 3Department of Metallic Materials Science, Physical Metallurgy, University Politehnica of Bucharest, 313 Splaiul Independentei, J Building, 060042 Bucharest, Romania

**Keywords:** LDH, chitosan, composite membranes, diclofenac, drug delivery

## Abstract

In this study, composite membranes based on chitosan (CS), layered double hydroxide (LDH), and diclofenac were prepared via dispersing of LDH and diclofenac (DCF) in the chitosan matrix for gradual delivery of diclofenac sodium. The effect of using LDH in composites was compared to chitosan loaded with diclofenac membrane. LDH was added in order to develop a system with a long release of diclofenac sodium, which is used in inflammatory conditions as an anti-inflammatory drug. The prepared composite membranes were characterized by Fourier Transform Infrared Spectroscopy (FT-IR), Scanning Electron Microscope Analysis (SEM), X-ray Photoelectron Spectroscopy (XPS), Thermogravimetric Analysis (TGA) and UV–Vis Spectroscopy. The results of the FTIR and XPS analyses confirmed the obtaining of the composite membrane and the efficient incorporation of diclofenac. It was observed that the addition of LDH can increase the thermal stability of the composite membrane and favors the gradual release of diclofenac, highlighted by UV–Vis spectra that showed a gradual release in the first 48 h. In conclusion, the composite membrane based on CS-LDH can be used in potential drug delivery application.

## 1. Introduction

The polymeric membranes have started to be used successfully in a very wide range of applications, such as in water treatment, gas separation, energy, electronics, and biomedicine [1,2,3]. Polymeric membranes have proven their efficiency and applicability for a wide range of biomedical processes, both for the separation of compounds of interest in the medical field as well as elements used in imaging [4], separation or degradation of antibiotics [5,6], hemodialysis [7], or tissue engineering [8,9]. The conventional methods of drug administration (such as through tablets administered orally or intravenous (IV) injection) present many disadvantages, such as the very high probability of side effects, the low solubility of the active substance, increased toxicity, the very low rapid drug activity, and rapid breakdown of the drug in vivo [10,11]. For these reasons, an attempt was made to develop controlled drug release systems that would solve these problems, so that the active substance would arrive in a targeted way and release gradually, so as to reduce side effects and systemic toxicity. Most polymeric membranes can be an ideal candidate in the development of systems for the controlled and targeted release of active substances due to their high porosity [12].

Chitosan (CS) is a linear polysaccharide with N-acetyl-D-glucosamine and D-glucosamine units, used in biomedical applications, due to its remarkable properties, such as its hydrophilicity, biodegradability, biocompatibility, and low cost [13,14,15,16]. Chitosan is a polysaccharide derived from chitin [17,18]. Chitosan is obtained through the deacetylation reaction of chitin isolated from the external skeleton and skin of arthropods and insects, and is the second most widespread polysaccharide, after cellulose [17,19,20]. However, chitosan, in addition to its high biocompatibility due to its biological nature, also has a high biodegradability, so it can be enzymatically degraded such that it can be broken down to obtain oligosaccharides and monosaccharides, and these molecules are able to be absorbed by the body very easily [21,22]. Chitosan-based materials are often used in drug delivery applications, such as oral drug delivery, cancer therapy, antibiotic drug delivery and gene delivery, due to their biocompatibility, biodegradability, low toxicity, and structural variability [22,23,24,25]. Further, CS is soluble in acidic media, so it can be used in various pH-sensitive controlled release systems [26,27,28]. In addition, many studies have shown that CS has antioxidant and antimicrobial properties [29,30,31,32,33,34]. Chitosan-based materials are versatile and can be obtained in different forms, such as films, hydrogels, fibers, and nano or microparticles [35].

Layered double hydroxide (LDH) is a group of inorganic materials made up of positively charged brucite-like layers with a charged interlayer [36]. The general formula of LDH materials is M^2+^ _(1-x)_ M^3+^_x_ (OH)_2_]^x+^ (A^n–^) _x/n_ · mH_2_O, where M^2+^ and M^3+^ are bivalent (Mg^2+^, Zn^2+^, or Ni^2+^) and trivalent metal cations (Al^3+^, Ga^3+^, Fe^3+^, or Mn^3+^), respectively. A^n-^ is the nonframework charge-compensating inorganic or organic anion, and x is the charge density of LDH layers and is obtained by the expression, x = [M^3+^]/([M^2+^]+[M^3+^] [37,38,39]. LDH-based materials have been successfully used as catalysts and anion exchangers, cement additives, fire retardant additives, carriers for drug release transporters in medicine, and antimicrobial biomaterials [37]. The lamellar structure with an increased surface and volume ratio of the LDH-based materials leads to the capability of incorporating large molecules [39]. The LDH-based materials are used in drug delivery systems as they incorporate the active substances between layers, and the drugs intercalated will be released slowly [40]. The major properties of LDH-based materials are high chemical stability, biocompatibility, and pH-dependent solubility, which could be used in the development of drug-delivery systems [41].

Inflammation is a normal response of the body to various stimuli including pathogens, damaged cells, and toxic compounds and can cause different symptoms, such as pain, fever and other complications [42]. In general, in the treatment of inflammation, various drugs with anti-inflammatory or analgesic effects are administered [43]. Diclofenac sodium is a nonsteroidal, anti-inflammatory analgesic agent generally used for the treatment of pain in the joint system [44]. The controlled release of diclofenac has been studied recently as diclofenac has the disadvantages of low aqueous solubility, low bioavailability, and short biological half-life [45,46,47]. Oral administration of diclofenac has been associated with the occurrence of several side effects, such as the occurrence of gastric ulcers, gastrointestinal bleeding, blood dyscrasias, and anaphylaxis [48]. Many researchers have tried different variants of controlled release of diclofenac in order to reduce the side effects associated with diclofenac administration [49,50,51,52].

For a gradual drug release from similar chitosan-LDH drug delivery systems, Barkhordari et al. [53] reported nanocomposite hydrogels based on chitosan/magnetic nanoparticles (Fe_3_O_4_ NPs) and Mg–Al double-layered hydroxides (LDH) in order to obtain pH-sensitive drug delivery systems for loading and the controlled release of diclofenac. The drug release is facilitated by the protonation of the functional groups of chitosan and the interlayer anions in LDH, leading to a high swelling of the samples at lower pH values. Lerner et al. [54] proposed chitosan/Mg/Al-NO3 layered double hydroxide (LDH) intercalated either with ibuprofenate anions (IBU) or a phospholipid bilayer (BL) containing a neutral drug. In this case, the release is triggered by the pH variation of the physiological environment, so that the release mechanism from the intercalated compounds occurs by the exchange between IBU and the buffer phosphate ions diffusing through the net of the polymer chains. Anirudhan et al. [55] reported a dual-drug delivery based on LDH/folic acid conjugated thiolated chitosan (TCS) and gold nanoparticle (AuNp) nanocomposite for the targeted therapy of breast cancer. The release of the antitumor agent occurs due to the pH difference at the tumor level and the photothermal stimuli.

The present study proposes the development of drug delivery systems for a gradual release of diclofenac, which were obtained based on composite membranes formed from chitosan and LDH. Diclofenac is loaded by the intercalation between the LDH layers and coated with chitosan to facilitate a gradual release, avoiding the rapid release of diclofenac in the first hours. The release mechanism occurs due to the ion exchange diffusing through the net of the polymer chains [54]. These composite membranes were evaluated from a morphological point of view through Scanning Electron Microscopy (SEM) and physico-chemical properties via Fourier Transform Infrared Spectroscopy (FTIR), X-ray Photoelectron Spectroscopy (XPS) and Thermogravimetric Analysis (TGA) in order to evaluate the possibility of use in biomedical applications such as controlled release systems.

## 2. Materials and Methods

### 2.1. Materials

Medium-molecular-weight chitosan (CS) was purchased from Sigma Aldrich (Saint Louis, MO, USA, with a deacetylation degree of 75% according to product technical sheet), LDH was synthesized according to the previously described method [56], acetic acid and diclofenac sodium was purchased from Sigma Aldrich (Saint Louis, MO, USA).

### 2.2. Preparation of Membranes

A chitosan (CS) solution was prepared by dissolving 2 g of chitosan powder in 18 mL of acetic acid for 30 min. at room temperature. Then, 180 mL of distilled water were added, and it was left to homogenize for 2 h at room temperature. The concentration of acetic acid solution used was 10 wt. %, and the concentration of CS solution was 1%. A volume of 10 mL of solution was poured into a Petri dish with a diameter of 10 cm and left to dry for 72 h at room temperature.

To obtain the membrane based on CS, loaded with DCF, we added diclofenac at a mass ratio CS:DCF of 1:0.05 to the previously obtained CS solution and left it to homogenize for 2 h, at room temperature. The obtained solution was poured into a Petri dish and left to dry for 72 h at room temperature.

0.5 g LDH were dispersed in 200 mL of distilled water for 30 min. at room temperature. After homogenization, 0.1 g of diclofenac was added and it was left to homogenize for another 2 h, also at room temperature. The obtained suspension was placed in Petri dishes and left to dry for 72 h at room temperature.

To obtain the composite membrane based on CS/LDH/DCF, LDH/DCF was added to the obtained CS solution, at a mass ratio CS:LDH:DCF of 1:0.25:0.05, and they were homogenized for 2 h at room temperature. A volume of 10 mL of solution was poured into Petri dish with a diameter of 10 cm and left to dry for 72 h at room temperature.

### 2.3. Characterization

Surface morphologies were evaluated using a XL 30 Field Emission ESEM (Philips, Amsterdam, The Netherlands) equipped with a high brightness field emission gun operating from 200 V to 30 kV with 20 Å resolution digital imaging. FTIR spectra were recorded on a Bruker VERTEX 70 spectrometer using 32 scans with a resolution of 4 cm^−1^ in 4000–600 cm^−1^ region. The samples were analyzed using ATR.

The surface chemistry was studied by X-ray Photoelectron Spectroscopy (XPS) using a K-Alpha instrument from Thermo Scientific (Thermo Fischer Scientific, Waltham, MA, USA), with a monochromated Al Kα source (1486.6 eV), at a bass pressure of 2 × 10^−9^ mbar. Charging effects were compensated by a flood gun and binding energies were calibrated by placing the C 1 s peak at 284.4 eV as internal standard. A pass energy of 200 eV and 20 eV were used for survey and high-resolution spectra acquisition respectively.

TGA curves were registered on a Q500 TA Instruments equipment, using nitrogen atmosphere, with a heating rate of 10 °C/min starting from room temperature (RT) to 800 °C.

The glass transition temperatures (Tg) of the composite membranes were established using differential scanning calorimetry (DSC). All the samples of around 10 mg were analyzed using a NETZSCH DSC 204 F1 Phoenix instrument (NETZSCH, Selb, Germany), under nitrogen flow, at 10 °C/min heating rate, in two heating/cooling cycles, between 18 and 350 °C.

The in vitro release of diclofenac was studied through a UV–Vis spectrophotometer (Shimadzu UV- 3600 UV–VIS). The samples were put into a dialysis membrane bag with 4 mL phosphate buffer solution (PBS) and then the samples were immersed in 200 mL PBS and spun for 72 h with 100 rpm at room temperature. The spectra were recorded at a maximum absorption wavelength of 276 nm and the standard curve was determined for concentrations of the drug between 2 and 10 µg/mL. The used membrane samples for release of diclofenac had 2 × 2 cm and a weight of 23 ± 3 mg. The measured standard curve is presented in Figure 1.

## 3. Results and Discussions

Morphological tests of the obtained membranes were performed using scanning electron microscopy (SEM). The observations were performed on the surface of CS-LDH membranes. Figure 2 shows SEM micrographs of the film surface at different magnifications (25× and 500×). In Figure 2A,B it can be observed that the neat chitosan has a smooth surface, without any porosity or roughness. After the addition of diclofenac, it can be seen that the diclofenac is unevenly dispersed, slightly agglomerated in the polymer matrix, possibly due to deficient magnetic mixing (Figure 2C,D). This could indicate the poor dispersion of diclofenac in chitosan solution. In Figure 2E,F can be seen a significant change in the morphology of the membrane, through the introduction of LDH, so that a slight stratification of the composite membrane is observed in comparison with the neat CS membrane (Figure 2A,B) and an increased dispersion of diclofenac in the polymer matrix. The addition of LDH is giving an additional protection against external degradation factors, but also for a more efficient, gradual release of the drug.

The cross-section images were captured by scanning electron microscopy (SEM) on CS, CS-DCF and CS-LDH-DCF composite membranes. The qualitative results are illustrated in Figure 3, which describes the morphology of each membrane on the effect of LDH addition. The neat CS membrane has a homogeneous structure, with variable thicknesses between 16 µm and 23 µm.

The cross-section SEM images for CS loaded with diclofenac shows a morphology similar to the cross-section of neat CS, which indicates that the addition of diclofenac does not change the morphology of the membrane, only a slight increase in thickness is observed, up to approximately 34 µm, It is also noticeable that in this case, too, we have an nonuniform thickness. The addition of LDH brought about a change in the morphology of the membrane, so that in the SEM cross-section image it can be observed that the structure of the membrane has become slightly stratified, confirming the assumptions deduced from the classic SEM images (Figure 2E,F). In addition, it can be observed that the thickness of the membrane increases due to the addition of LDH, so that a variable thickness between 36.2 µm and 48.1 µm resulted.

The information provided by morpho-compositional SEM analyses were confirmed by XPS. The results are presented in Figure 4 and in Table 1. For the sample of neat chitosan, the following characteristic elements were identified: O1s (17.46%), C1s (75.92%) and N1s (3.58%).

In addition to the characteristic elements, silicon was also identified, possibly due to the impurity of the sample. After the addition of diclofenac, there is a slight increase in atomic resolution for C1s (75.95%), O1s (18.32%), and Na1s (5.08%), which is a characteristic element of diclofenac, indicating the dispersion of diclofenac in the matrix of chitosan. The addition of LDH is represented by the appearance of characteristic elements, such as S2p (4.09%), Si2p (1.87%), and Al2p (3.1%). For the composite membrane formed by CS-LDF-DCF, all the constituent chemical elements of the composite membrane were highlighted, so that O1s (24.35%), C2s (65.78%), N1s (3.16%), S2p (0.93%), Si2 (2.95%), and Al2p (1.89%), indicating the presence of both LDH and diclofenac encapsulation.

The FTIR was carried out to highlight the obtaining of the composite membrane based on CS, LDH, and DCF. The FTIR spectra for the samples for CS, LDH, DCF, CS-DCF, LDH-DCF, and CS-LDH-DCF are shown in Figure 5. The characteristic peaks for chitosan were observed in all the samples in which the chitosan membrane was used, and, as a result, the following characteristic peaks were observed: at 1030 cm^−1^, which was attributed to the stretching vibration of C-O in C-O-C groups [57,58]; 1070 cm^−1^, attributed to the C-O stretching bands [59]; 1155 cm^−1^, attributed to the stretching vibration of C-O in C-O-H groups [57]; 1550–1560 cm^−1^ corresponds to the amino group in chitosan [60]; 1641 cm^−1^ corresponds to N-H deformation bending of chitosan [61]; and the 2354–2356 cm^−1^ peaks correspond to the presence of C–H stretching vibrations, which is indicative of the presence of methyl groups [62]. The characteristic peaks that highlight the presence of LDH are 1646 cm^−1^, characteristic of the carbon–carbon double bond (C=C) [56]; at 1373–1386 cm^−1^, attributed to the interlayer NO_3_ anion groups [53]; between 1010 and 1100 cm^−1^, characteristic of the Si–O bond; and between 630 and 640 cm^−1^, which were related to metal–oxide bonds that indicate Mg–O and Al–O [53,63] being present, both in the case of the curve of the LDH-DCF sample and in the curve of the CS-LDH-DCF sample. The presence of the drug diclofenac is noted by the appearance of bands characteristic at 1453 cm^−1^ and 747 cm^−1^, and are associated with CH_2_CH_3_ deformation and C–Cl stretching [64], which can be observed for the CS-DCF sample as well as for the LDH-DCF and CS-LDH-DCF samples. The presence of all these peaks at CS-LDH-DCF composite membranes not only confirms the successful preparation of it, but also represents the effective drug loading.

The thermal stability of the as-synthesized membranes was studied through Thermogravimetric Analysis (TGA) under nitrogen atmosphere. In Figure 6 are presented the TGA and DTG curves of neat CS, CS loaded with diclofenac, and the composite membrane based on CS-LDH-DCF. According to the figure, CS and CS-DFC membranes show similar decomposition profiles with two degradation steps. The first step, between 50 and 130 °C, was due to the evaporation of interlay water molecules, and the second step, between 160 and 400 °C, was attributed to the degradation of the amino/N-acetyl groups from the CS chain [65,66,67,68]. Moreover, in the case of CS-LDH-DCF composite membrane, a new small weight change, between 400 and 520 °C, could be observed and was due to the degradation of the interlayer anionic species and also to the dehydroxylation of the brucite-like structure of LDH [54,69]. Furthermore, an increase in the residual mass loss in the case of the CS-LDH-DFC composite membranes has also been recorded. The temperature at which the mass loss is 50% (Td_50%_) was also recorded, and an enhancement of about 54 °C (from 321 °C to 375 °C) could be observed in the case of the CS-LDH-DFC composite membranes compared with pure CS membrane. This significant increase was attributed, on the one hand, to the good dispersion and intercalation of the nanofiller within the polymer matrix and, on the other hand, to the good interfacial interaction between the nanofiller and polymer matrix [70]. This leads to an increase in the thermostability of the composite membranes.

The TGA analysis was further supported by DSC analysis where the DSC thermograms were displayed in Figure 7. The DSC curve of pure CS showed an exothermic peak at about 275 °C, attributed to the polymer chain degradation, including saccharide ring dehydration, depolymerization, and decomposition of deacetylated and acetylated chitosan unit, and an endothermic peak at about 75 °C associated with the evaporation of absorbed water [71,72]. In the case of CS-DCF, these peaks were moved to a lower temperature compared to the pure CS, and this is due to the fact that the addition of diclofenac within CS matrix slightly decreased the thermal resistance of the membranes. However, after a close examination, we can also observed the presence of a new, slightly broad endothermic peak around 195 °C into CS-DFC curve which could be attributed to the breaking of the hydrogen bonds obtained between CS and DFC. This has been also observed into the FTIR dates where a shift of the N–H bond (1643 cm^−1^) to a higher value was registered in CS-DFC sample. Conversely, in the case of CS-LDH-DCF composite membranes, both peaks at 275 °C and 75 °C were shifted to a higher value and confirm the TGA results, indicating that the addition of LDH within the CS matrix increases the thermostability of the composite membranes.

The tests for the release of diclofenac from the composite membrane were carried out by the UV–Vis technique according to the procedure developed in a previous research [73] at pH 7.4 and the data are presented in Figure 8.

The tests were carried for 72 h; in the first 6 h, 26% of the drug quantity was released. In the first 24 h, 50% was released; in 48 h, 88% was released; and in 72 h 90% was released for system CS-DCF. In the first 6 h, 16% was released; in the first 24 h, 31% was released; in 48 h, 46% was released; and in 72 h, 65% was released for system CS-LDH-DCF. There are several mechanisms that lead to the delivery of drugs from a polymer matrix, the most important being related to diffusion-controlled release, the swelling-controlled release, erosion and degradation-controlled release, and stimuli-controlled release [74]. Diffusion-controlled released appears in the case of porous polymer matrix and is governed by the diffusion mechanism [74,75]. Swelling-controlled released appears when polymer chains break down under the interactions that occur between polymer and the surrounding media, a process that is preceded by swelling [74,76]. Polymer erosion involves swelling, diffusion, and dissolution, the drug release being regulated by the type of polymer, and the internal bonding of other species present in the system [74,75]. Stimuli-controlled release occurs when the shape, size, porosity, or other properties of the polymer matrix are affected by an external stimuli such as pH, temperature, or the presence of biological markers, and the drug is released based on these interactions, this being the most advantageous strategy in terms of delivery to the destination and minimizing side effects [74,77,78]. In the case of chitosan-based drug delivery systems, the release mechanism is a combination of all these mechanisms depending on the synthesis method and the surrounding media of the polymer system. Chitosan compact films synthesized for drug delivery in strong acidic pH (1.5 for gastric juice) or moderate acidic pH (5.5–6.5 for cancer tumors) are systems with fast release capacity due to the polymer structure [22]. In such systems, especially those designed for release in strong acidic media, the entire quantity of active pharmaceutical substance is released very quickly, in several hours. Due to the insolubility of chitosan at normal extracellular pH (6.8–7.4), the release is much slower and occurs because of the formation of a hydrogel [79]. In our case, the highest release for CS-DCF system is based on this gel structure formation, while in the case of the CS-LDH-DCF system, despite the fact the film forms the same gel structure, the release occurs from the layers of LDH by extracting the diclofenac in the media, a process that takes longer. Similar results were obtained for release of sodium diclofenac from system Chitosan-Oxidized Konjac Glucomannan by Korkiatithaweechai et al. [80]. The obtained data show an ideal efficiency for the first 48 h after a potential implantation, and the release profile shows that the obtained membrane can provide an efficient anti-inflammatory response for the critical period, with applications in stomatology and orthopedy.

## 4. Conclusions

In summary, we developed a composite membrane-based CS-LDH for diclofenac drug release via solvent evaporation. The SEM micrographs showed a change in the surface morphology, so that the neat CS membrane had a smooth surface, and the CS-LDH-DCF composite membrane surface has a rough architecture, due to the addition of LDH. The SEM images demonstrated the successful incorporation of LDH by changing the morphology of surface and thickness of the membrane. The FTIR and XPS analyses demonstrated the efficient loading of diclofenac in the CS-LDH-DCF composite membrane. The TGA and DSC results showed a slight increase in thermal stability after the addition of LDH. The drug release study indicates a gradual release in the critical period, being capable of effective anti-inflammatory response. Therefore, the combination design of the CS-LDH-DCF composite membrane provided a reserving strategy for drug delivery systems in inflammatory diseases.

## Figures and Tables

**Figure 1 membranes-13-00179-f001:**
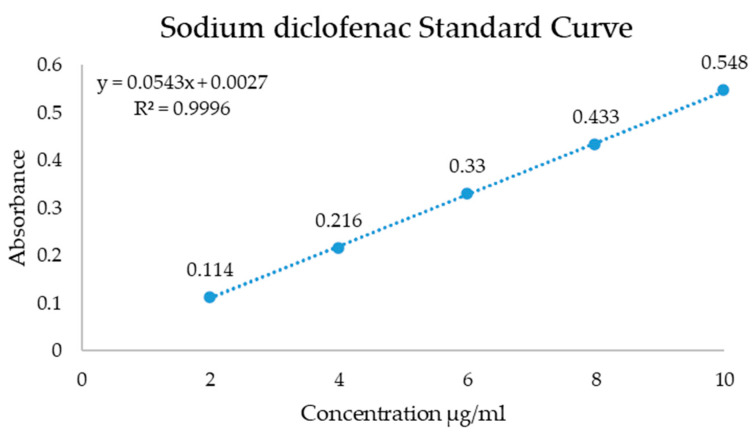
Standard curve for determination of diclofenac release from membranes.

**Figure 2 membranes-13-00179-f002:**
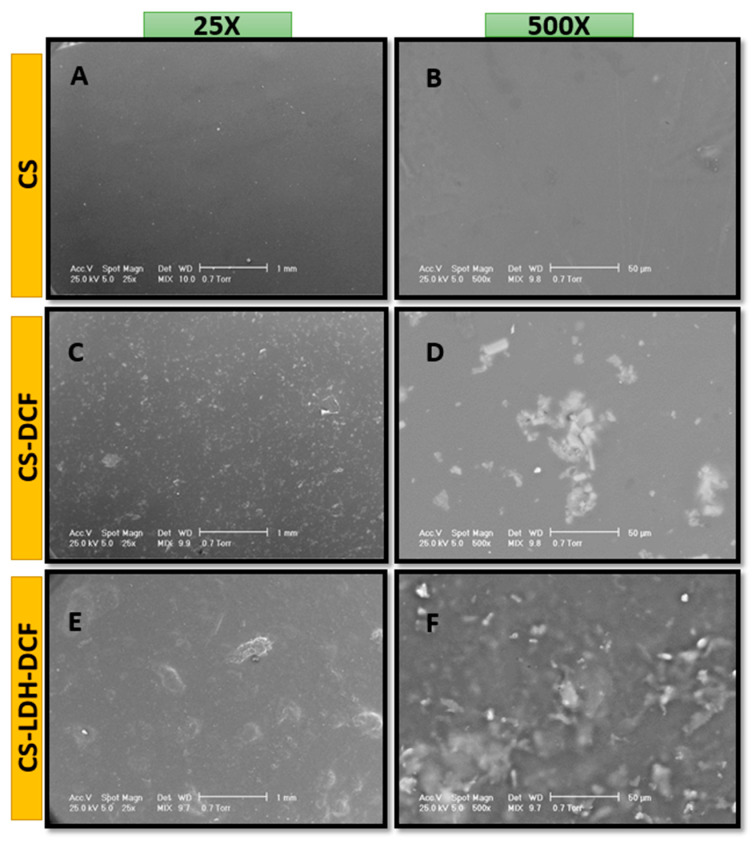
SEM images of neat chitosan (CS, (**A**)—25×, (**B**)—500×), chitosan loaded with diclofenac (CS-DCF, (**C**)—25×, (**D**)—500×), and chitosan–LDH loaded with diclofenac (CS-LDH-DCF, (**E**)—25×, (**F**)—500×).

**Figure 3 membranes-13-00179-f003:**
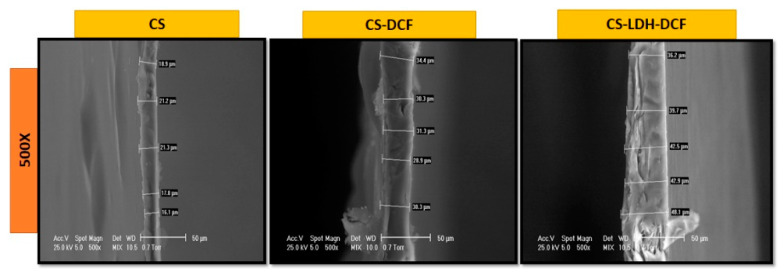
SEM images of the cross-section view of chitosan, chitosan loaded with diclofenac, and chitosan–LDH loaded with diclofenac.

**Figure 4 membranes-13-00179-f004:**
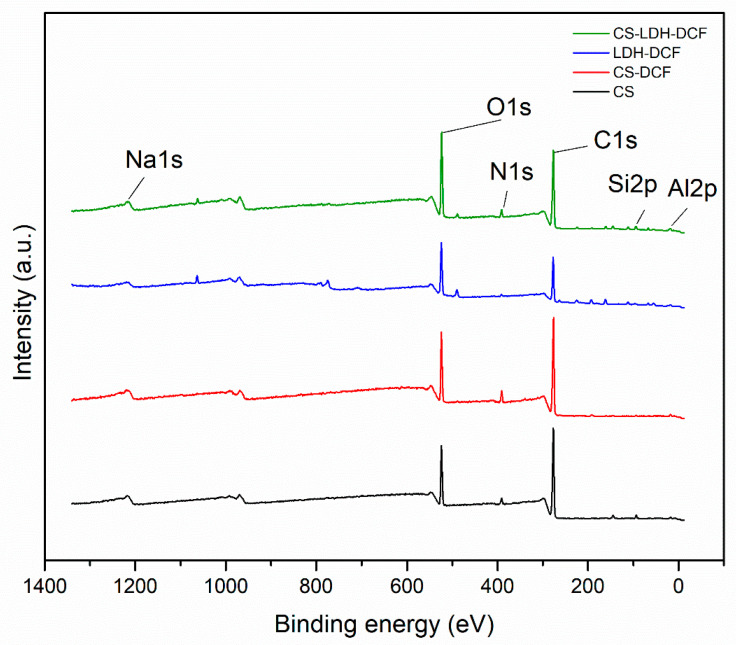
X-ray photoelectron spectroscopy (XPS) of CS, CS-DCF, LDH-DCF and CS-LDH-DCF.

**Figure 5 membranes-13-00179-f005:**
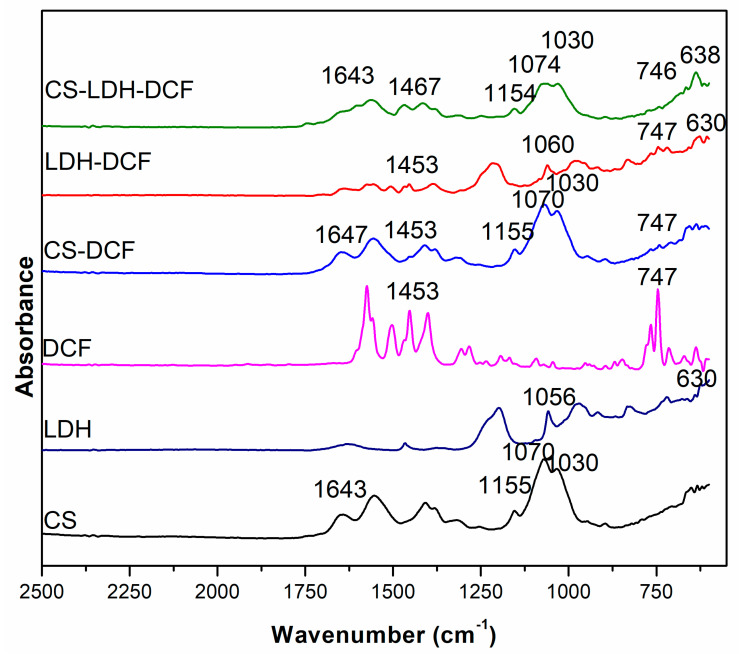
FTIR spectra for CS, LDH, DCF, CS-DCF, LDH-DCF, and CS-LDH-DCF.

**Figure 6 membranes-13-00179-f006:**
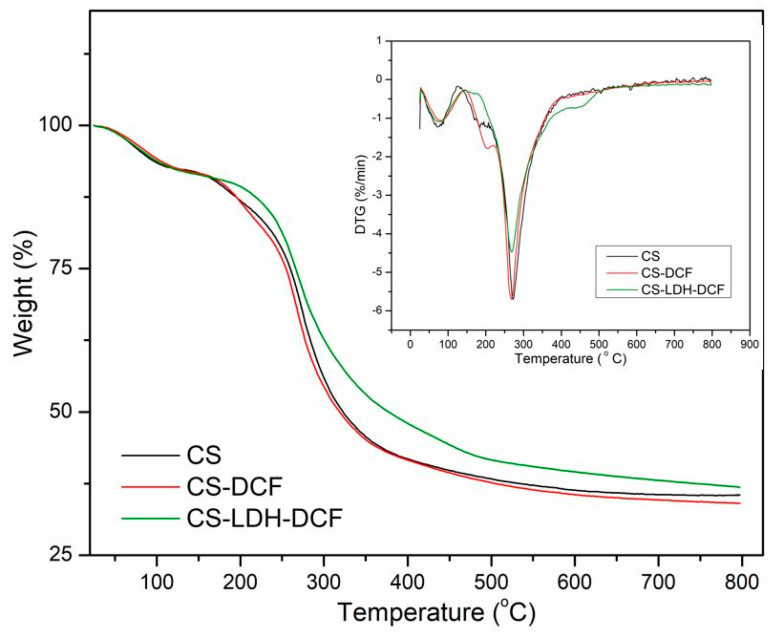
Thermal analysis of neat CS, CS-DCF, and CS-LDH-DCF.

**Figure 7 membranes-13-00179-f007:**
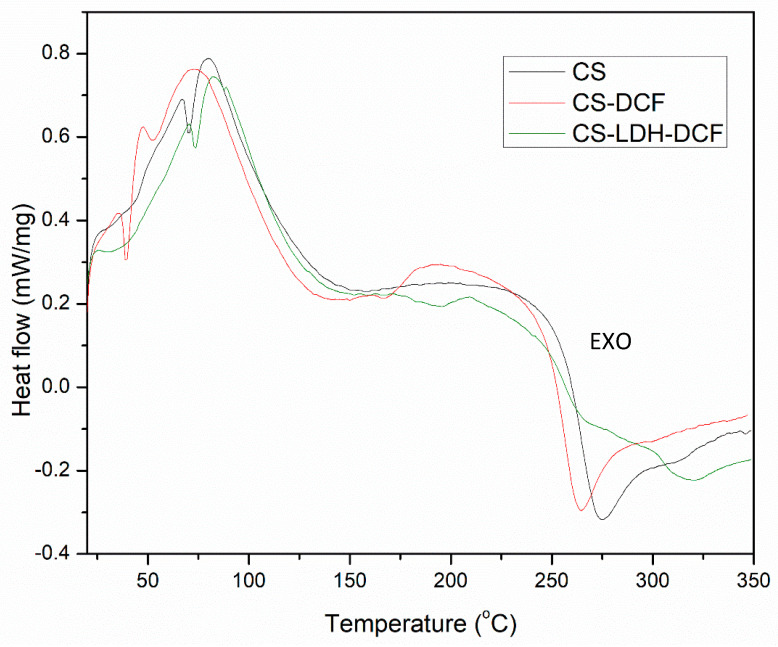
Differential scanning calorimetry (DSC) analysis of the neat CS, CS-DCF, and CS-LDH-DCF.

**Figure 8 membranes-13-00179-f008:**
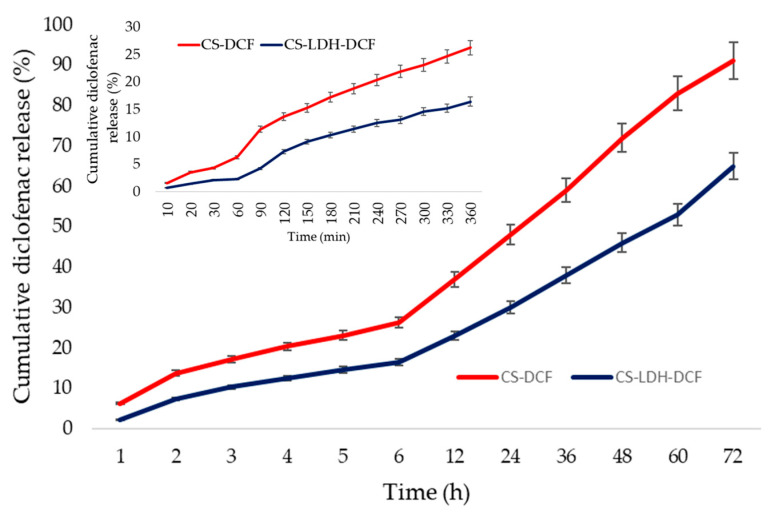
Cumulative diclofenac (%) release in time (h) for first 72h and inset (min) for first 6 h.

**Table 1 membranes-13-00179-t001:** XPS elemental data analysis.

Sample	Atomic Percent (%)						
	O1s	C1s	N1s	Na1s	S2p	Si2p	Al2p
**CS**	17.46	75.92	3.58	-	-	2.79	-
**CS-DCF**	18.32	75.95	0.65	5.08	-	-	-
**LDH-DCF**	25.69	56.71	1.79	2.33	4.09	1.87	3.1
**CS-LDH-DCF**	24.35	65.78	3.16	0.96	0.93	2.95	1.89

## Data Availability

Raw data are available upon request at corresponding author.

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
