# Peer review of "Preparation and Characterization of Chitosan/LDH Composite Membranes for Drug Delivery Application"

_membranes, 2023, doi:10.3390/membranes13020179_

Round 1

Reviewer 1 Report (Previous Reviewer 2)

The present form of the manuscript is acceptable but Figure 7 needs to be redrawn because Figure 7 is not publication standard so needs to be redrawn with high quality then only readers can understand.

Author Response

Thank you for your time and effort for evaluation of our manuscript. Please see attachment.

Reviewer 2 Report (Previous Reviewer 1)

The authors have revised the manuscript, so it could be accepted in the present form.

Author Response

Thank you for your time and effort for evaluation of our manuscript.

Reviewer 3 Report (New Reviewer)

General comments:

No information about the possible application of chitosan-based materials in drug delivery systems was found in the introduction. Why has CS been chosen for this research project?

The introduction looks like being composed of several definitions (of Cs, LDH, and others). The introduction does not explain the problem that needs to be solved and why the CS, LDH etc, have been applied in the drug-release system's formation.

Why was the controlled release testing performed only in PBS? It is well known that CS is a polyelectrolyte and thus release rate is highly dependent on the external pH, especially in the acidic pH (i.e. below ca. 6.5). If the material, as given in the title, is devoted to drug delivery application the release experiments should be also performed in other external conditions. Moreover, the description of DCF release methodology is laconic.

Other comments:

Line 12: explain all abbreviations when they are used for the first time (chitosan, LDH)

Lines 38-40: not all polymeric membranes exhibit high porosity. A lot of drug delivery systems are based on dense, non-porous membranes.

Line 44: “N-acetyl-d-glucosamine” should be “N-acetyl-D-glucosamine”

Line 45: “with one amino (NH2) group and two hydroxyl (OH) groups in each repeating glycosidic unit,” – it is not true as some of the units are not deacetylated; thus they do not have amino groups

Line 48: “Chitin is extracted through the deacetylation reaction from the external skeleton” – not true; deacetylation is a process of transformation of chitin into chitosan (see i.e., https://www.ncbi.nlm.nih.gov/pmc/articles/PMC4377977/)

Paragraph 2.1: characterization of chitosan, especially its deacetylation degree, should be given

Line 117: “2.2 Preparation of Chitosan (CS)” – this paragraph describes the preparation of CS solution, not CS itself

Line 119: concentration of acetic acid solution should be given

There is no info about the final concentration of CS solution, the mass ratio between CS (in a solution) and DCF (see 2.3/2.5), or volume:mass ratio between chitosan solution and DCF (see 2.3/2.5). It is necessary to know the initial loading of diclofenac [per gram of the polymer]

Line 157: was the nanocomposite structure confirmed (TEM, XRD?) If not, such a statement is overestimated.

Line 176: is it really diclofenac agglomerates? How was it confirmed (EDX?) Why the DCF was not well dispersed – please add more discussion of these results

Lines 181-183: this sentence is “overestimated” – the interaction of DCF with LDH and the material's molecular structure can not be confirmed with the SEM images alone.

Line 190: “The morphology of the obtained membranes presents an asymmetric structure.” Explain – indicate the layers of asymmetric material in figure 2

The statement about membrane thickness does not correspond to the data provided in Fig. 2 (e.g. 1.6-2.3 um for CS membrane, while in figure 2, values between 16.1 and 21.3 um can be seen). Not only give the results but discuss them as well.

Figure 2 and 3: must be corrected – only important/informative wavelengths should be shown (without 3000-1800) to get better resolution, and some essential bands should also be marked with values. Moreover, figs. 2 and 3 should be shown as one as some of the spectra are repeated (CS, CS-LDH-DCF

Paragraph 3.2: As Author stated: “The FTIR was carried out to highlight the interaction between chitosan, LDH and Diclofenac.” – but there is no confirmation in the text indicating the possible interactions between all components. It is evident that if we mixed 3 ingredients, the characteristic vibration bands of all of them could be found in the spectra. What are the possible interaction between CS, LDH, and DCF – it is crucial as these interactions can affect the release phenomenon.

XPS analysis: the elemental analysis will be better if it is coupled with SEM to confirm the distribution of DCF in the polymeric matrix. XPS mapping of elements being a component of additives like Mg/Si will only improve the data and analysis. Moreover, some of the statements of diclofenac distribution in XPS and SEM discussion do not correspond to each other.

Lines 245-247: it should be stressed that the XPS analysis represents only the film surface. Moreover, in XPS analysis -  the decrease in Na content is not an indication of the less diclofenac content, especially when another component is added (i.e CS-DCF and CS-LDH-DCF). The representative is the ratio between Na content and the standard element.

Line 273-274: once Author writes that all samples show similar mass losses – but in my opinion, only in the first thermal event. Moreover, it was assumed that due to the addition of LDH the mass lost is observed in the 350-550 range while the mass lost is also observed within this range for CS and CD-DCF samples.

Line 286: these are not spectra but thermograms

Lines 288-300: the DSC thermal events at 220 at 275 are mostly related to polymer degradation. It can also be seen in TG data, where the high mass loss occurs in this same region.

Diclofenac sodium release is not well discussed. Moreover, there are mistakes suggesting the existence of pores inside the membrane, while these results were not confirmed with SEM images. What means “ideal efficiency”? No release rate was shown and compared.

The conclusions are simply a repetition of the result discussion. It should be rewritten and corrected.

The English language needs extensive editing

Other mistakes: “pf”, “Figure 2 which describes the morphology”

Round 2

Reviewer 3 Report (New Reviewer)

I have read all the answers and corrections carefully and still have some issues to be addressed:

Line 41 – should be high instead of increased

Line 46 – should be obtained instead of extracted

Line 52- change "very" into "often"

Line 200: remove "due to the incorporation of diclofenac" L

Line 201: instead of "inhomogenous" it is better to write "non-uniform"

Line 258-260 – not only elimination but rather the degradation of chitosan and then elimination of the low molecular products of this degradation process

The whole TG and DSC discussion needs to be revised:

Line 260: in my opinion, it is visible that the degradation of CS and CS-DCF is similar (similar temperatures of the thermal events and corresponding weight losses) but CS-LDH-DCF exhibits lower weight changes in the thermal degradation event- Authors also indicated this in the following sentence.

Line 262-267 – style of this sentence needs to be revised

Line 273-275 – In my opinion, there are only two thermal events of CS – the effect at 220 C.deg is almost invisible (1st – evaporation of residual water, 2nd – degradation (not evaporation) at 275). In the DSC curve of CS-DCF there is a new peak at 195 – please check if this is not the temperature of DCF evaporation (e.g., 10.1023/A:1025421911670)

Line 300 – in the manuscript, there is no proof regarding the membrane's dense/porous structure. What's more, the cross-sections of the films indicate dense, non-porous structure (commonly seen for chitosan films obtained by the evaporation technique)

As the aim of this work was focused on the preparation of membranes with prolonged DCF release, the discussion of this release should be improved and refer to the structural characteristic of the membranes.

Line 313 – there was no info in the manuscript text regarding the layered structure in the cross-section. What does this layered structure mean?

Round 3

Reviewer 3 Report (New Reviewer)

The manuscript has been improved according to my previous suggestions and can be accepted in the present form.

This manuscript is a resubmission of an earlier submission. The following is a list of the peer review reports and author responses from that submission.

Round 1

Reviewer 1 Report

The manuscript describes composite membranes based on chitosan, LDH and diclofenac were prepared via dispersing of LDH and diclofenac in the chitosan matrix for gradual delivery of diclofenac sodium. The results indicated the successful inclusion of LDH and diclofenac in the chitosan matrix for potential drug delivery application.

Specific comments

1. Why did the author use “acetic acid” to dissolve chitosan (rather than acetic acid dilute aqueous solution) ? Chitosan is easier to degrade at higher acid concentration.

2. I suggest the authors provide the fracture surface (cross-section) morphologies of the composite films, which is better to understand the distribution of LDH in the chitosan membrane? 

3. The figure caption of Figure 1 and the corresponding description should be revised, and the letters in the figure 1 were not consistent with those in the description (A-F vs a-f).

4. Figure 2, Figure 3 and Figure 4 should be revised.

5. Why did the maximum rate of decomposition temperatures (Tmax) decrease with the incorporation of LDH?

6. The current conclusion is poor, please provide some data in the conclusion.

Author Response

Thank you for time and effort in evaluating our manuscript, please see attachment.

Reviewer 2 Report

  1. Insert the FTIR graphs of diclofenac and LDH, and it will give clarity to the readers regarding the interactions between polymer, drug, and LDH.
  2. Check the axis of the FTIR graph.
  3. How can you conclude that the drug has dispersed in the matrix based only on FTIR results? The FTIR results are not sufficient to discuss the interaction of the drug with the polymer matrix. Perform the studies of XRD and DSC; these studies support your statement.
  4. I am wondering how the membrane shows sustained release. Please give a reference for this statement. The Fig. 5 is not publication standard. You conducted the studies over 72 hours, why the graph shows 24 hours only.
  5. The developed membrane is pH-responsive? Discuss it. What are the applications of developed membranes, whether they are used orally or by any other route?
  6. Discuss the mechanism in the discussion section to see why it releases the drug slowly and give references.
  7. Provide the encapsulation efficiency results of the developed membranes.
  8. The abstract is not publication standard; improve it and the English language also.

Author Response

(The authors gave the same response as above.)
